# InstructRestore: Region-Customized Image Restoration with Human Instructions

**Shuaizheng Liu**[1,2]**, Jianqi Ma**[1]**, Lingchen Sun**[1,2]**, Xiangtao Kong**[1,2]**, Lei Zhang**[1,2,†]

[1]The Hong Kong Polytechnic University    [2]OPPO Research Institute

shuaizhengliu21@gmail.com, {jianqi.ma, ling-chen.sun, xiangtao.kong}@connect.polyu.hk
cslzhang@comp.polyu.edu.hk

## Abstract

Despite the significant progress in diffusion prior-based image restoration for real-world scenarios, most existing methods apply uniform processing to the entire image, lacking the capability to perform region-customized image restoration according to user preferences. In this work, we propose a new framework, namely **InstructRestore**, to perform region-adjustable image restoration following human instructions. To achieve this, we first develop a data generation engine to produce training triplets, each consisting of a high-quality image, the target region description, and the corresponding region mask. With this engine and careful data screening, we construct a comprehensive dataset comprising 536,945 triplets to support the training and evaluation of this task. We then examine how to integrate the low-quality image features under the ControlNet architecture to adjust the degree of image details enhancement. Consequently, we develop a ControlNet-like model to identify the target region and allocate different integration scales to the target and surrounding regions, enabling region-customized image restoration that aligns with user instructions. Experimental results demonstrate that our proposed InstructRestore approach enables effective human-instructed image restoration, including restoration with controllable bokeh blur effects and region-specific restoration with continuous intensity control. Our work advances the investigation of interactive image restoration and enhancement techniques. Data, code, and models are publicly available at `https://github.com/shuaizhengliu/InstructRestore.git`.

## 1   Introduction

Image restoration (IR) is a fundamental problem in computer vision to recover high-quality images from degraded inputs. Early works have achieved significant progress on individual IR tasks based on specific simulated degradation assumptions, including denoising [59, 60], deblurring [28, 36], and super-resolution [9, 24]. While demonstrating strong performance within their target domains, these approaches exhibit inherent limitations when generalizing to real-world scenarios characterized by unknown and composite degradations. This has motivated the emerging paradigm of real-world image restoration, which aims to handle complex degradation in practical imaging scenarios, particularly for challenging cases like real-world super-resolution [5, 43]. To address this challenge, recent advances have developed sophisticated degradation models to better approximate real-world conditions [57, 39]. Building upon these advanced degradation models, and with the advent of pretrained text-to-image (T2I) generation models such as Stable Diffusion (SD) [33], which can more effectively model the complex distribution of natural images, researchers have started to explore the use of powerful SD priors to produce realistic IR outcomes [38, 25, 50, 46, 52, 45, 35, 41, 2, 8, 51].

---

† Corresponding author. This work is supported by the PolyU-OPPO Joint Innovative Research Center.

39th Conference on Neural Information Processing Systems (NeurIPS 2025).

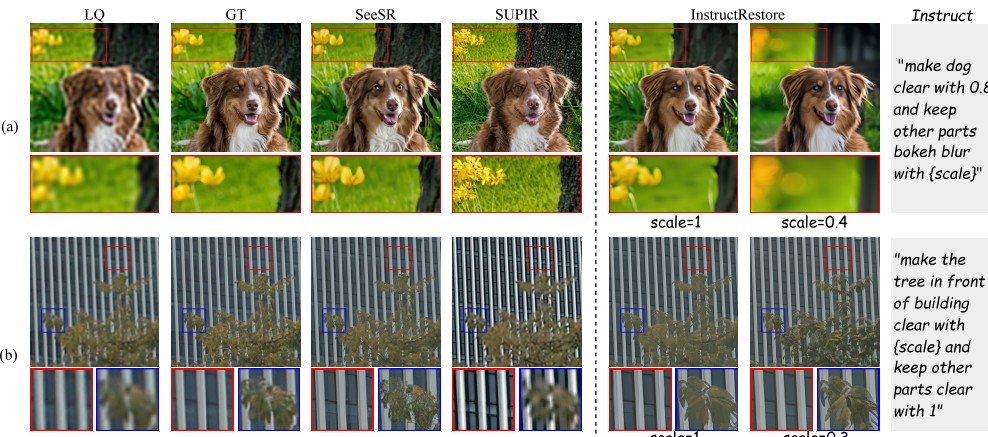

Figure 1: Our proposed **InstructionRestore** framework enables region-customized restoration following human instruction. As shown in (a), current methods [46, 52] tend to restore the bokeh blurry region incorrectly, while our approach allows for adjustable control over the degree of blur based on user instructions. In (b), existing methods fail to achieve region-specific enhancement intensities, while our approach can simultaneously suppress the over-enhancement in areas of building and improve the visual quality in areas of leaves.

By generating details semantically consistent with the underlying content of the image, SD-based methods achieve significantly better perceptual quality than previous approaches [23, 26, 6, 9, 34, 29, 40, 39, 22]. However, image restoration is an ill-posed problem that admits multiple plausible solutions. Existing methods are limited to a single restoration outcome applied uniformly across the image, lacking the ability to accommodate varied user preferences across different image regions. For example, in the photography of targeted objects (*e.g.*, portrait), the background is intentionally blurred for aesthetic focus. During restoration, users typically want to preserve or even adjust bokeh effects, yet existing generative prior-based IR methods may produce unnecessary textures that disrupt the intended bokeh effects on the background regions, as shown in Fig. 1(a). In addition, user preferences for content fidelity and perceptual quality vary across image semantic regions. For irregular texture regions (*e.g.*, trees), it's challenging to accurately recover pixel-wise details due to signal aliasing in the degradation process [22]. In these cases, strictly enforcing fidelity often leads to over-smoothed results. Therefore, users generally prioritize perceptual quality for irregular texture regions, favoring more aggressive detail generation. Conversely, for structural regions (*e.g.*, architecture) or flat areas (*e.g.*, skies), large pixel-wise differences are more perceptually sensitive and easily detected as artifacts [47]. Therefore, user preference may shift towards content fidelity to preserve accuracy, as illustrated in Fig. 1(b). Unfortunately, existing methods cannot achieve such customized restoration of different regions.

To address the limitation mentioned above, we propose **InstructRestore**, a novel framework that enables users to realize region-specific restoration through natural language instruction for real-world scenarios, including bokeh adjustment and region-aware tuning of content fidelity and perceptual quality. Our InstructRestore approach can precisely adjust restoration effects in target semantic regions while keeping other areas unaffected, showing the ability of instruction following. To begin with this novel task, we need a dataset for training and evaluation, which should offer descriptions of target regions to construct human instruction, along with corresponding region masks. To the best of our knowledge, there is not a publicly available dataset that provides such triplets of high-quality images, referring descriptions, and the corresponding region masks. The most relevant datasets to our task can be the referential segmentation datasets such as RefCOCO [53]. However, its image quality and resolution are insufficient to support IR tasks. To bridge this gap, we develop a data generation engine. Utilizing Semantic-Sam [18] and Osprey [55] models, we obtain masks and initial descriptions from a set of selected high-quality images. We then use large language models (LLMs), more specifically Qwen [48], to iteratively parse and refine these descriptions, formatting them to meet the instructional requirements of IR tasks. Finally, we build a dataset of 536,945 triplets, covering diverse scenes such as plants, buildings, animals, *etc*.

Building upon this dataset, we train the InstructRestore model for region-customized IR with user instructions. To ensure that the model can accurately identify the human-specified region and properly

enhance the designated area, we propose integrating the conditional features of low-quality input images into a ControlNet-like architecture. Instructions are used as text prompts to the control-branch of ControlNet [61]. Trained on our curated dataset, the control-branch could simultaneously generate region masks and conditional features. By applying distinct integration scales to the conditional features of user-customized regions and their surroundings, our InstructRestore model achieves locally controlled restoration that aligns with user intentions.

Our key contributions are summarized as follows. (1) First, we introduce the task of region-customized image restoration with human instruction, which represents an important class of practical IR tasks. (2) Second, we develop a data generation engine and construct a large-scale dataset with $536, 945$ triplets to support this task. (3) Finally, we present InstructRestore, the first model that understands user instructions for region-customized restoration for real-world complex degradation. Our experiments demonstrate the capability and effectiveness of our InstructRestore model, showcasing its great potential for interactive and user-instructed image restoration.

## 2   Related Work

**Diffusion-based Restoration in Real World**. Recent diffusion models have significantly advanced the task of IR in real world, addressing mixed degradations such as noise, blur, JPEG compression, and resolution reduction. StableSR [38] and DiffBIR [25] treat the low-quality (LQ) input as condition to guide reverse diffusion process. PASD [50] and SeeSR [46] introduce the semantic prompts like short captions or tags to enrich the result with finer semantic details. SUPIR [52] scales up datasets along with long descriptions to boost perceptual quality with SDXL [30] pre-trained model. DreamClear [2] and FluxIR [8] introduce Diffusion Transformer (DiT)-based models designed for enhanced performance in image restoration. To tackle the inefficiency of iterative sampling, one-step diffusion methods [45, 35, 56] have emerged, ensuring quality with faster inference. Despite their advancements, existing methods perform restoration uniformly, failing to accommodate user preferences for region-specific refinements.

**Instruction-guided Editing and Restoration**. Natural language instructions enable intuitive human-AI collaboration by translating high-level intent into pixel-level operations. Instruction-guided image editing methods like InstructPix2Pix [4] and MagicBrush [58] have demonstrated remarkable capabilities in spatially aware manipulations. Subsequent works like MGIE [10] and SmartEdit [13] further advance instruction comprehension through multimodal LLMs. Others [19, 11] focus on region-specific control, ensuring editing explicitly defined areas by user instructions. However, these breakthroughs remain confined to semantic-level manipulation rather than physically grounded restoration. To address this, recent efforts have incorporated user instructions into restoration frameworks. InstructIR [7] and PromptFix [54] leverage task-specific instructions to enable a single model to handle multiple restoration tasks, including denoising, deblur, rain removal, *etc*. SPIRE [31] incorporates semantic descriptions to handle in-the-wild restoration scenarios. However, these methods primarily use instructions for task differentiation or global parameter tuning, lacking the ability to perform region-specific refinements. Our work introduces the first instruction-guided restoration framework for real-world scenarios that enables region-specific refinements through natural language commands, addressing the critical limitation of global-only operations in prior arts.

## 3   Dataset Construction

InstructRestore aims to adjust restoration effects on user-specified regions following human instructions. To achieve this goal, the model needs to understand the semantic information of the target regions for performing localized restoration. A critical requirement for training such a model is the availability of a large-scale dataset, which simultaneously offers high-quality images, descriptions of target regions, and corresponding region masks. In this paper, we develop a data generation engine to build such a comprehensive dataset, named **Tri-IR**, to facilitate the research of InstructRestore tasks. The data generation process is detailed in Fig. 2.

### 3.1   Dataset Construction Pipeline

**High-quality ground-truth image collection.** High-resolution and high-quality ground-truth (GT) images are critical for training IR models. Therefore, we collect high-quality images from LSDIR [20],

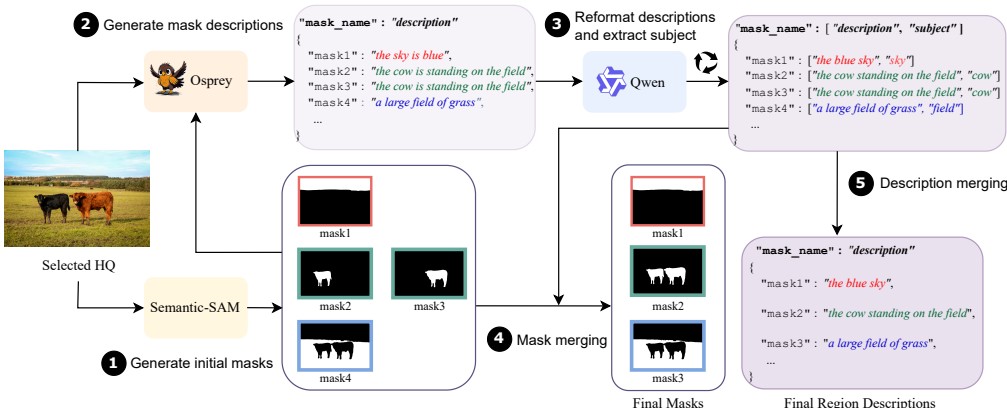

Figure 2: Illustration of the annotation pipeline. For selected high-quality images, Semantic-SAM [18] generates initial masks, followed by Osprey [55] for region-level descriptions. Qwen [48] reformats descriptions into noun phrases and extracts semantic subjects. Identical semantics are merged to produce final masks and region captions.

EntitySeg [32] train set and EBB! [14] bokeh train set with shorter side larger than 512 pixels and MUSIQ [17] score larger than 60.

**Annotation pipeline.** To obtain high-quality images, we design an automatic annotation pipeline to extract the semantic region masks and their corresponding descriptions with a combination of state-of-the-art models. In the mask extraction phase, we first utilize a state-of-the-art segmentation model, *e.g.*, Semantic-SAM [18], to generate coarse-grained semantic segmentation masks for the semantic region of the images. For images from EntitySeg [32], we directly reuse their pre-annotated masks. Once the masks are obtained, we pair each image with its mask and feed them into a multi-modal large language model, Osprey [55], to generate region-level descriptions. These descriptions serve as part of instructions to specify the regions to be processed or restored. At this stage, though we obtain preliminary masks and descriptions, they are still far from perfect as our training data due to two key issues below: (1) Semantic-SAM [18] occasionally produces multiple mask pieces for one semantic meaning, leading region ambiguity and harmful for the region customization learning; (2) the descriptions are not always in noun phrase format due to the response arbitrariness, making them unsuitable for embedding into instructions.

To address these issues, we first utilize Qwen-7B [3], a large language model (LLM), to perform the following tasks through prompt tuning: (1) parsing the subject from the descriptions and (2) reformatting them into noun phrases. Due to the randomness in LLM's outputs, we iteratively perform the refinement process. Specifically, we identify error cases and re-execute the above process by a larger LLM, Qwen-72B [3]. This cycle is repeated 3 times to ensure high-quality outputs. More details can be found in the **appendices**. Finally, based on the parsed subject, we merge the masks and their corresponding descriptions for regions with identical semantics.

## 3.2 Dataset Statistics

As shown in Fig. 2, our Tri-IR dataset, provides triplets of high-quality GT images, region masks, and descriptive captions. To underscore the relevance and utility of our dataset, we compare it with the most relevant referential segmentation datasets including RefClef [16], RefCOCO [53], RefCOCO+ [53] and RefCOCOg [27] in Table 1, which also provide masks and captions for semantic regions. The comparison focuses on the number of annotations, the range of image resolutions, and MUSIQ-based quality scores.

Table 1: Statistics of our dataset and related datasets.

| Datasets | Annotation Amount | Min Resolution | Max Resolution | MUSIQ |
|---|---|---|---|---|
| RefClef [16] | 99,523 | 320×480 | 360×480 | 67.06 |
| RefCOCO [53] | 196,771 | 157×160 | 640×637 | 69.73 |
| RefCOCO+ [53] | 196,737 | 157×160 | 640×637 | 69.73 |
| RefCOCOg [27] | 208,960 | 157×160 | 640×637 | 69.73 |
| Ours | 536,945 | 540×540 | 4464×2244 | 71.87 |

As can be seen from Table 1, existing datasets, while widely used for segmentation, exhibit critical limitations for IR tasks. Their images are capped at resolution less than 650 pixels and their MUSIQ scores fall significantly below ours. In contrast, our dataset not only provides 536,945 annotated regions (surpassing other datasets in scale) but also delivers higher-resolution images with superior

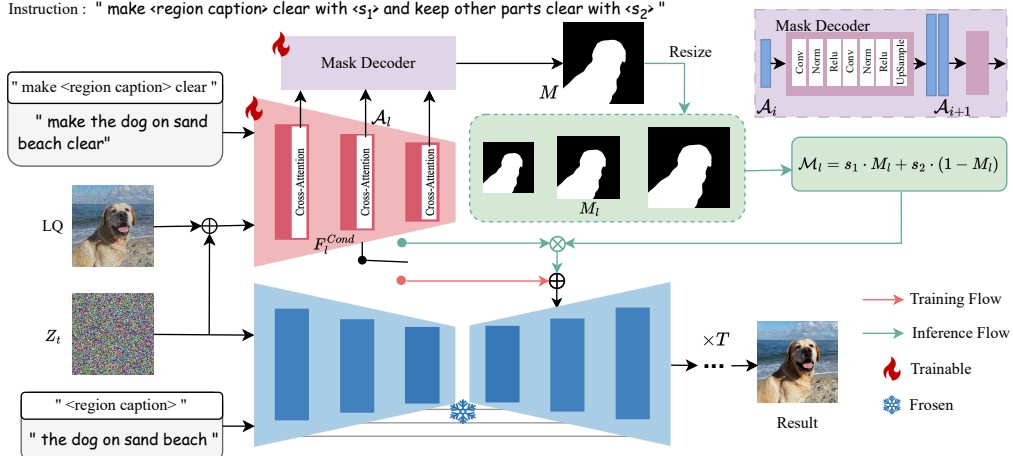

Figure 3: Framework of InsturctRestore. The framework uses red and green arrows to denote training and inference processes respectively. During testing, user instructions are parsed to generate target-region semantic masks, with differentiated coefficient modulation applied to conditional features inside/outside mask regions, enabling instruction-guided region-specific restoration effects.

perceptual quality, meeting the need for IR tasks. Our dataset enables both precise semantic control and photorealistic restoration. To further illustrate the semantic diversity and applicability of our dataset, we plot a word cloud reflecting the relative frequency of different semantic content in the **appendices**, which demonstrates that our dataset covers a wide range of semantic regions that are commonly targeted in restoration tasks, such as plants, buildings, animals, *etc*.

# 4 InstructRestore Model Design

Our network is designed to achieve region-specific restoration effects based on user instructions, where each instruction contains both spatial region specifications and restoration strength. The core challenge lies in how to accurately localize specified regions and implement continuous controllable local restoration effects with given strength, while keeping the restoration of remaining regions unchanged. A straightforward approach might be to construct training data pairs corresponding to different local restoration strengths. However, this approach is too labor-intensive. We develop a more elegant solution to address the above challenges without requiring different strength pairs. Specifically, existing SD-based IR models often employ a ControlNet architecture with the low-quality (LQ) image as a conditional signal. In this architecture, the pre-trained SD backbone generates text-guided features while the ControlNet branch extracts LR-derived features. The fusion between these two pathways determines the final restoration output. We observe that scaling the ControlNet features by a coefficient $\alpha$ during inference provides flexible control of the data fidelity and semantic enhancement. Intuitively, a smaller coefficient allows the SD backbone to dominate, resulting in richer generated details and enhanced perceptual quality, while a larger coefficient makes the output closer to the degraded image, increasing fidelity but reducing generated details. Building on this insight, we develop **InstructRestore**, which employs a ControlNet-like architecture during training using only standard restoration data. The network learns to perform restoration while predicting region masks from user instructions. During inference, InstructRestore generates region masks and scales ControlNet feature by different coefficients inside and outside the masks based on user instructions to achieve region-specific restoration.

## 4.1 Training Framework

**Architecture design**. As shown in Fig. 3, our InstructRestore model consists of a pre-trained SD backbone, the ControlNet adaptor, and a lightweight mask decoder. The SD model is frozen during the entire training stage. The region captions $c_R$ extracted from the user instructions $c_I$ act as text prompts for the SD model, providing semantic guidance to generate semantic details. ControlNet duplicates the encoder and middle blocks of the SD UNet as trainable copies. It receives features

extracted from the LQ image and user instructions $c_I$ as input, then extracts hierarchical conditional features from the input and injects them into the UNet decoder blocks at multiple scales.

To accurately localize the target regions in user instructions, we design a mask decoder to predict a spatial mask $\hat{M}$. Since ControlNet is initialized from the pre-trained SD UNet, it has been revealed [12] that the cross-attention features between textual and visual embeddings exhibit strong responses to text-described semantic regions. We then extract cross-attention features $\{\mathcal{A}_l\}_{l=1}^{L}$ between textual embeddings and visual features at each scale of the ControlNet as input to the Mask decoder, which is designed with a pyramidal structure to effectively process multi-scale features. The features of each scale $\mathcal{A}_l$ are first passed through two blocks, each consisting of a convolutional layer (Conv), group normalization (GN), and a ReLU activation. The processed features are then upsampled and concatenated with the cross-attention features $\mathcal{A}_{l+1}$ at the next scale. The combined features are processed by another Conv-GN-ReLU block and passed to the subsequent scale.

**Training process**. Our constructed dataset consist of triplets $[I_{\text{HQ}}, M, c_M]$, where $I_{\text{HQ}}$ denotes the high-quality GT image, $M$ is the binary mask specifying the target region, and $c_M$ is the textual caption of the masked area. To generate training samples, we first apply the Real-ESRGAN degradation pipeline to $I_{\text{HQ}}$ to obtain the LQ input $I_{\text{LQ}}$. Subsequently, we construct specific instructions $c_I$ and region caption $c_R$ based on $c_M$, tailored to different restoration purposes. For region-specific restoration, $c_I$ follows the template *"make $\{c_M\}$ clear"*, while $c_R$ is the same as $c_M$. For bokeh-aware restoration, the template becomes *"make $\{c_M\}$ clear and keep other parts bokeh blur"*, while $c_R$ follows the template *"$\{c_M\}$ in front of bokeh background"*.

During training, $I_{\text{HQ}}$ is first encoded into the latent space by the pre-trained VAE encoder, yielding $z_0$. The diffusion process progressively corrupts $z_0$ with Gaussian noise over randomly sampled timesteps $t$, resulting in noisy latent states $z_t = \sqrt{\alpha_t} z_0 + \sqrt{1 - \alpha_t} \epsilon$, where $\epsilon \sim \mathcal{N}(0, \mathbf{I})$ and $\alpha_t$ follow a cosine noise schedule. We utilize $z_t$ and region caption $c_R$ as inputs to pre-trained SD backbone. The ControlNet takes $I_{\text{LQ}}$, $z_t$ and $c_I$ as input to produce conditional features $\{F_l^{\text{cond}}\}_{l=1}^{L}$, which are added to the frozen SD UNet decoder with scaling factor $\alpha = 1$ during training. The training flow is highlighted in red in Fig. 3.

The mask decoder takes the cross-attention features $\{\mathcal{A}_l\}_{l=1}^{L}$ from ControlNet as input to generate target region masks $\hat{M}$, supervised by the GT masks $M$ with Cross-Entropy loss. The InstructRestore network, denoted by $\epsilon_\theta$, is conditioned on noisy latent $z_t$, LQ images $I_{\text{LQ}}$, instruction $c_I$, and region caption $c_R$. The training objective $\mathcal{L}$ combines the diffusion loss and mask supervision $\mathcal{L}_{\text{mask}}$:

$$\mathcal{L} = \mathbb{E}_{t,\epsilon} \left[ \|\epsilon - \epsilon_\theta(z_t, t, I_{\text{LQ}}, c_I, c_R)\|_2^2 \right] + \lambda \mathcal{L}_{\text{mask}}(\hat{M}, M), \tag{1}$$

where $\mathcal{L}_{\text{mask}}(\hat{M}, M) = \text{CrossEntropy}(\hat{M}, M)$ and $\lambda$ balances the two terms.

## 4.2 Region-customized Inference

After training, our framework enables users to specify target regions and restoration intensities through structured instructions during inference, as shown in the green flow in Fig. 3. The user instructions follow task-specific templates. For general restoration, we set the template as *"make {region caption} clear with $\{s_1\}$, and make other parts clear with $\{s_2\}$"*; for bokeh-aware restoration, the template is *"make {region caption} clear with $\{s_1\}$, and keep other parts bokeh blur with $\{s_2\}$"*. The {region caption} specifies the textual caption of region of interest (*e.g.*, "the dog on the sand beach"), and $s_1, s_2 \in \mathbb{R}^+$ define the enhancement scales towards fidelity for the target and other regions, respectively. Here, larger values of $s_1$ and $s_2$ result in higher fidelity (closer to the degraded input), while smaller values allow more semantic enhancement. The instruction parsing process extracts three key components: the region caption for SD backbone text conditioning, the main instruction body for ControlNet text encoding, and the fidelity scales $s_1, s_2$ for mask modulation. For region-customized restoration, the region caption is directly used as SD text condition, while for bokeh-aware restoration, it is modified to *"{region caption} in front of bokeh background"* to stimulate the pre-trained SD backbone to generate bokeh blur effect features. Similarly, the main instruction body differs between tasks: *"make {region caption} clear"* for general restoration and *"make {region caption} clear and keep other parts bokeh blur"* for bokeh-aware restoration.

The trained ControlNet branch processes the degraded image and the parsed instruction to generate a mask $M \in [0, 1]$ indicating the target region. It is dynamically resized to match the spatial dimensions of each U-Net upsampling decoder layer, producing masks at multiple scales $\{M_l\}_{l=1}^{L}$. At each layer

$l$, a modulation map is computed as:

$$\mathcal{M}_l = s_1 \cdot M_l + s_2 \cdot (1 - M_l), \tag{2}$$

where $s_1$ and $s_2$ control the fidelity scales for the target and background regions, respectively. This modulation map determines how much ControlNet features contribute to the final output: higher values preserve more original content, while lower values allow more semantic enhancement. The modulated ControlNet features $F_l^{\text{cond}}$ are fused with the base SD features $F_l^{\text{sd}}$ via element-wise multiplication:

$$F_l^{\text{out}} = F_l^{\text{sd}} + \mathcal{M}_l \odot F_l^{\text{cond}}. \tag{3}$$

By applying this modulation progressively across all decoder layers, our framework ensures precise alignment with user intent, *i.e.*, enhancing target regions with intensity $s_1$ while maintaining natural fidelity in non-target areas with intensity $s_2$. Our architecture seamlessly transitions between differently enhanced regions, producing photorealistic restoration results that follow user instructions.

## 5 Experiments

### 5.1 Experiment Settings

**Training details.** Our method is built on SD2.1 [33]. Training data is generated by the data generation engine described in Section 3. The LQ images are obtained by the Real-ESRGAN [39] degradation pipeline. The LQ images and instructions serve as inputs to the model, while the GT images and region masks provide supervision. Our model is first trained on the general degradation dataset for 120K iterations, guided by the instruction template *"make the { region caption } clear "*. The training continues by combining the bokeh dataset with the general degradation dataset for 14k iterations. During this stage, the sampling probability is set to $25\%$ for the general degradation dataset and $75\%$ for the bokeh dataset, which is paired with the instruction template *"make the { region caption } clear and keep other parts bokeh blur."*. The training is conducted on two A100 GPUs with a batch size of 64 and an initial learning rate of $5e^{-5}$. AdamW is adopted as the optimizer for network training.

**Comparison methods.** As the first instruction-based region-customized IR approach, InstructRestore mainly benchmarks against: (1) GAN-based Real-ESRGAN [39]; (2) Diffusion-based methods like StableSR [38], DiffBIR [25], PASD [50], SeeSR [46], SUPIR [52], and OSEDiff [44].

### 5.2 Results on Localized Enhancement

We first show InstructRestore's results with user instructions. Then we demonstrate its precise restoration of specified regions. Finally, we compare it with existing methods.

**Test dataset**. We curate 100 real-world images from multiple sources, including RealSR [5], DRealSR [43], and the RAIM challenge [21], to construct our Instruct100Set. Specifically, we select and crop images with clear semantic region to ensure meaningful evaluation. The foreground masks are generated using the pipeline described in Section 3, ensuring accurate and consistent ROI extraction. The user instructions used in the experiment are in the format of *"make { target region caption } clear with { fidelity level 1 } and keep other parts clear with { fidelity level 2 }."*

**Evaluation metrics.** To comprehensively assess the performance of our method, we adopt both reference-based and no-reference metrics, evaluating both target regions and the entire image. The reference-based metrics include PSNR, SSIM [42] (on the Y channel in YCbCr space) and LPIPS [62]. The no-reference metrics include MANIQA [49], MUSIQ [17] and CLIPIQA [37]. For region-specific evaluation, we compute PSNR and SSIM exclusively within human-specified regions by using the provided GT mask. For other metrics, we zero out pixels outside the target region based on GT mask for computation. This ensures the evaluation focusing on the target regions while being compatible with standard implementations of these metrics.

**Localized enhancement with user-instruction**. We first showcase our method's ability to perform localized enhancement with user-instructions. By specifying the target region and enhancement strength, our method allows users to explicitly control the balance between data fidelity and generative details. As illustrated in Fig. 4, by applying different fidelity scale instructions to the flower region, we successfully adjust the level of details in the flower region while keeping other regions (*e.g.*, leaves and soil) largely unchanged. To our best knowledge, our method is the first one to allow user-instructed local enhancement. To quantitatively validate the instruction-following capability

Table 2: Quantitative evaluation on the instruction following capability of InstructRestore. Experiments are conducted on the Instruct100Set with instruction of "make { region caption of target area} clear with { fidelity scale } and keep other parts clear with 1."

| Fidelity Scale | Target Area | | | | | | Remaining Area | |
|---|---|---|---|---|---|---|---|---|
| | PSNR↑ | SSIM↑ | LPIPS↓ | CLIPIQA↑ | MUSIQ↑ | MANIQA↑ | PSNR↑ | SSIM↑ |
| 0.5 | 29.71 | 0.7522 | 0.1610 | 0.6801 | 67.86 | 0.6108 | 31.27 | 0.8949 |
| 0.7 | 30.37 | 0.8188 | 0.1439 | 0.6931 | 68.23 | 0.6161 | 31.55 | 0.9047 |
| 0.9 | 30.64 | 0.8494 | 0.1331 | 0.6832 | 67.91 | 0.6091 | 31.61 | 0.9087 |
| 1.1 | 30.73 | 0.8649 | 0.1253 | 0.6659 | 66.92 | 0.5934 | 31.56 | 0.9108 |

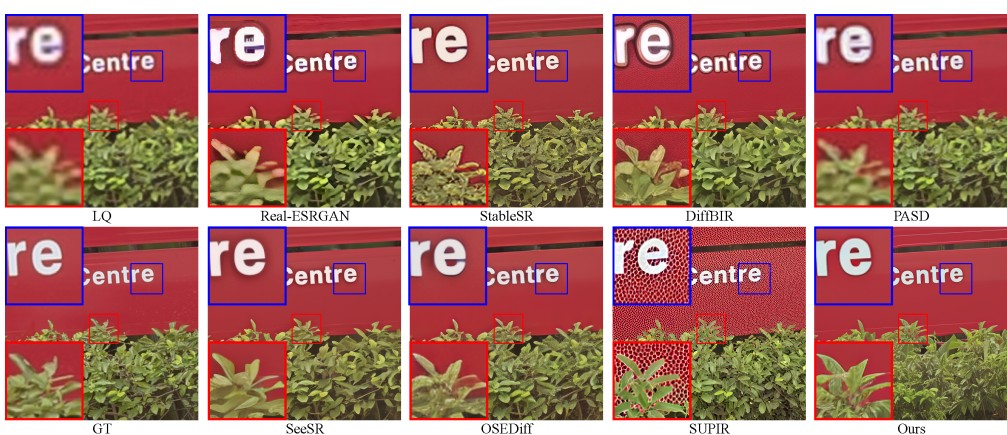

Figure 5: Visual comparison of different methods. We set the instruction as *"make the bush in front of sign clear with 0.5 and keep other parts clear with 0.9"* to keep the fidelity of sign and prioritize detail enhancement of bushes.

of our method, we conduct experiments by varying the enhancement fidelity scales exclusively within the target region while keeping the fidelity scale of the surrounding areas unchanged. For the surrounding regions, we calculate reference-based metrics to assess their stability. The quantitative results are shown in Table 2. We see that when the fidelity scale is small, the non-reference metrics for the target region are significantly higher, indicating that the method tends to generate more details.

As the fidelity scale increases, the reference-based metrics (*e.g.*, PSNR and SSIM) improve, while the non-reference metrics gradually decrease. This demonstrates that the method effectively follows the instructions, transitioning from detail-oriented generation to a more input-faithful reconstruction. Furthermore, the PSNR of the target region varies by $1.02$ db, while the surrounding regions vary only by $0.29$ db. This stark contrast confirms that the enhancement process is localized to the target region, leaving the surrounding areas largely unaffected. Due to space limitations, ablation studies on the mask decoder and feature modulation mechanism are provided in the **appendices**. We also conduct instruction variation experiments in the appendix, testing with instructions that do not follow the standard templates. Although not trained on such variations, our model demonstrates reasonable robustness in generating masks. Please refer to the **appendices** for details.

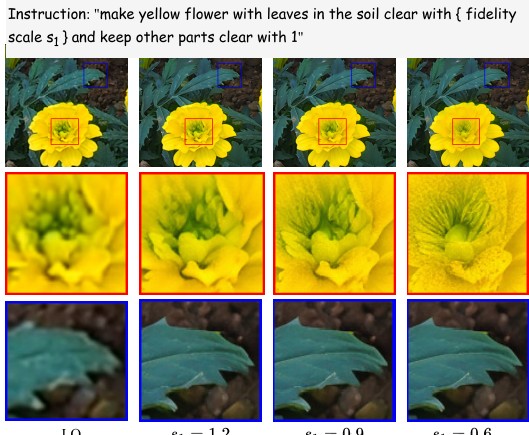

Figure 4: Localized enhancement following instruction on real-world test data. The details in flowers are enhanced gradually while the other regions keeping almost unchanged.

**Comparison with other methods**. We then compare InstructRestore with the competing methods. For images with heavier degradations, we prioritize stronger generative prior to synthesize more details; for regions with high-frequency and irregular textures (*e.g.*, flowers, brushes), we favor generative enhancement to achieve realistic appearances; while for regions with regular structures (*e.g.*, sign and buildings), a conservative enhancement level is selected to avoid unnatural artifacts. As shown in Figure 5, our method can handle well distinct regions, namely the sign and the bushes,

Table 3: Quantitative comparison between our InstructRestore method and other methods on Instruct100Set. The best and second best results of each metric are highlighted in red and blue.

| Method | Target Area | | | | | Full Image | | | | | |
|--------|------|------|---------|--------|---------|------|------|-------|---------|--------|---------|
| | PSNR↑ | SSIM↑ | CLIPIQA↑ | MUSIQ↑ | MANIQA↑ | PSNR↑ | SSIM↑ | LPIPS↓ | CLIPIQA↑ | MUSIQ↑ | MANIQA↑ |
| RealESRGAN | 31.69 | 0.9065 | 0.7124 | 58.63 | 0.4991 | 27.69 | 0.7871 | 0.3185 | 0.7280 | 60.39 | 0.5030 |
| StableSR | 30.36 | 0.8522 | 0.6707 | 65.75 | 0.5915 | 25.39 | 0.7072 | 0.3001 | 0.7072 | 69.19 | 0.6691 |
| DiffBIR | 30.95 | 0.8804 | 0.6820 | 66.80 | 0.5971 | 26.64 | 0.6897 | 0.3434 | 0.7456 | 69.96 | 0.6609 |
| PASD | 31.80 | 0.9176 | 0.5724 | 61.02 | 0.5323 | 28.37 | 0.7893 | 0.2590 | 0.5768 | 62.92 | 0.5866 |
| SeeSR | 30.90 | 0.8788 | 0.6758 | 67.73 | 0.5974 | 26.75 | 0.7324 | 0.2879 | 0.7246 | 71.49 | 0.6691 |
| SUPIR | 30.74 | 0.8682 | 0.6868 | 62.98 | 0.5655 | 26.29 | 0.6997 | 0.3235 | 0.6840 | 64.40 | 0.6085 |
| OSEDiff | 30.21 | 0.8657 | 0.6417 | 66.75 | 0.5851 | 26.07 | 0.7340 | 0.2870 | 0.7342 | 71.88 | 0.6635 |
| Ours | 30.55 | 0.8368 | 0.6887 | 68.17 | 0.6137 | 25.65 | 0.6999 | 0.3245 | 0.7278 | 71.95 | 0.6809 |

within the same scene. In comparison, methods such as DiffBIR and SUPIR tend to over-enhance the sign, introducing unnecessary artifacts and distortions, while other methods fail to adequately reconstruct the bush, resulting in a smeared and over-smoothed appearance.

Our method allows adjusting the fidelity scale to meet the specific requirements of each region. For example, for the sign, which requires high fidelity, we set the fidelity scale to $0.9$ for faithful restoration. For the bush, we prioritize detail enhancement with a fidelity scale of $0.5$ to generate richer textures. To provide an example of quantitative evaluation, we simply set the fidelity scale for the foreground at $0.8$, while that for other regions to $1$. The evaluation results are reported in Table 3. Since this setting prioritizes generative enhancement in target regions to achieve richer details, it shows better no-reference metrics but relatively lower scores in reference metrics that favor strict fidelity preservation. It is important to note that our InstructRestore enables users to adaptively adjust restoration results based on their preferences. The metrics here only serve as an example to demonstrate that our approach can produce visually pleasing results following user instructions.

## 5.3 Results on Images with Bokeh Effects

In this section, we perform experiments to demonstrate that our method can perform image restoration while preserving bokeh effects and controlling the bokeh intensity.

**Test dataset**. We construct a test dataset by selecting images from two sources: the EBB! dataset [14] and images with bokeh background carefully curated from Pixabay [1]. We select $70$ images from them with distinct semantic foregrounds and bokeh background. Masks are generated for the foreground regions to precisely define the ROI, based on which the images are center-cropped to ensure a consistent resolution of $512 \times 512$. Subsequently, we apply Real-ESRGAN [39] degradations to generate LQ and GT image pairs for evaluation. To support instructed interaction, we leverage the masks and GT images to generate foreground descriptions using the pipeline illustrated in Section 3. The user instructions are formatted as *"make {foreground description} clear with {enhancement fidelity level} and keep other parts bokeh blur with the {bokeh level}."*

**Evaluation Metrics.** We compute reference-based metrics for full image and background regions. In addition, we employ D-DFFNet [15], a model for detecting blurred background, to generate the background mask and compute the Intersection-over-Union (IoU) with GT of background mask as a measure of bokeh preservation performance.

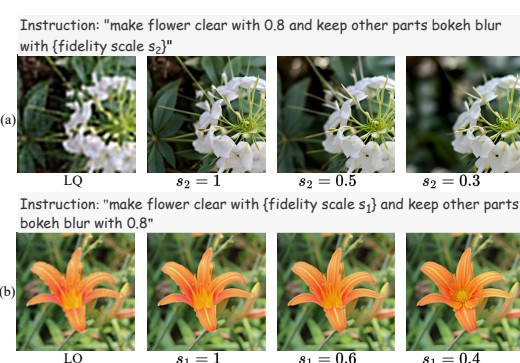

Figure 6: Control of bokeh effect and foreground enhancement. (a) Restoration with controlled bokeh effect while restoring foreground. (b) Restoration with varying foreground enhancement levels while preserving background bokeh.

**Control of bokeh effect**. Our method allows users to specify the desired intensity of bokeh effects and foreground enhancement level via instructions. As illustrated in Fig. 6 (a), our method successfully adjusts the background blur based on user instructions, simulating varying depth-of-field effects while maintaining the sharpness and details in the foreground. More importantly, the adjusted blur is not merely a uniform increase in blur intensity. It faithfully replicates the circular light spots of realistic bokeh, mimicking the optical effects produced by high-quality digital single-lens reflex (DSLR) cameras. As mentioned in Section 4.2, we stimulate the

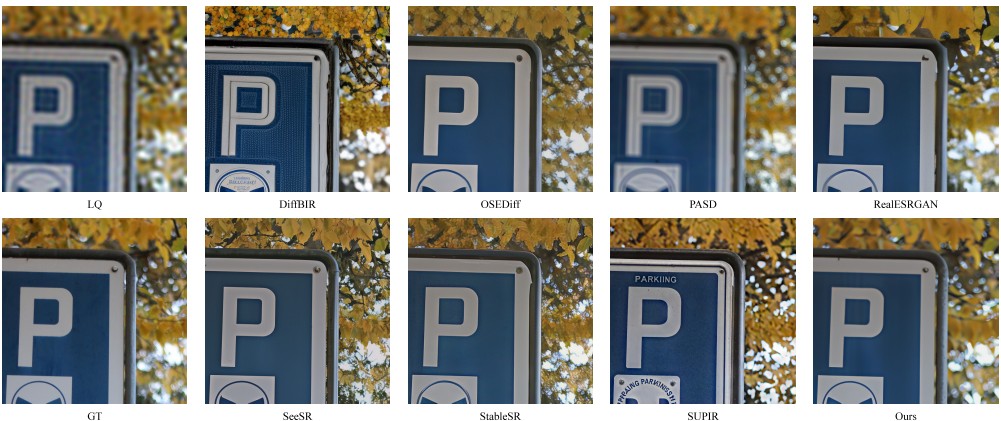

Figure 7: Visual Comparison of bokeh preservation results between the compared methods.

generation of authentic bokeh features by incorporating "bokeh blur background" text into the SD backbone. So smaller scale values result in less integration of LR input features, leading to increased blur effects, enabling simulation of different depth-of-field effects. In addition to specifying the intensity of bokeh effects, users can further specify the enhancement strength for the foreground, achieving flexible control on the level of details in focal regions. As illustrated in Fig. 6 (b), the fine details in the flower stamens become more pronounced as the instruction changes. Note that such a feature is not supported by existing restoration methods. We also provide quantitative results validating our controllable blur adjustment in the **appendices**.

**Comparison with other methods**. Since the foreground semantics in EBB! mainly include objects like cars and road signs requiring high fidelity, we set the foreground enhancement strength to 1.0. For simplicity and fairness, the bokeh fidelity scale is also set to a default value of 1, representing the weakest depth-of-field effect. The quantitative comparison results are presented in Tab. 4.

Our method demonstrates significantly better performance in fidelity-oriented metrics compared to competing methods, reflecting its ability to accurately approximate the GT's bokeh characteristics. In contrast, existing methods fail to preserve bokeh effects, leading to deviations from the GT. Visual comparison is shown in Fig. 7. More comparisons are provided in the **appendices**. Competing methods tend to restore

Table 4: Quantitative comparison on Bokeh testset

| Method | Background | | | Full Image | | |
|---|---|---|---|---|---|---|
| | PSNR↑ | SSIM↑ | Bokeh IoU↑ | PSNR↑ | SSIM↑ | LPIPS↓ |
| RealESRGAN | 30.86 | 0.8305 | 0.7203 | 23.69 | 0.7060 | 0.3700 |
| StableSR | 30.24 | 0.8049 | 0.7405 | 22.55 | 0.6305 | 0.3965 |
| DiffBIR | 30.46 | 0.8017 | 0.6289 | 22.20 | 0.5943 | 0.4415 |
| PASD | 31.87 | 0.8453 | 0.8234 | 24.27 | 0.7280 | 0.3523 |
| SeeSR | 30.42 | 0.8149 | 0.7580 | 22.95 | 0.6652 | 0.3677 |
| SUPIR | 29.92 | 0.7847 | 0.7739 | 21.21 | 0.5745 | 0.4375 |
| OSEDiff | 29.89 | 0.8175 | 0.7990 | 22.58 | 0.6707 | 0.3609 |
| Ours | 31.46 | 0.8462 | 0.8482 | 24.69 | 0.7437 | 0.3394 |

the background with sharp details, disrupting the bokeh effect, whereas our method preserves natural background blur while enhancing foreground details, ensuring both fidelity and artistic quality.

## 6 Conclusion

We presented InstructRestore, the first framework for region-customized image restoration guided by human instructions. To support this task, we designed a scalable data annotation engine and constructed a dedicated dataset comprising 536,945 triplets, each containing a high-quality image, the region mask and region caption. Building on this dataset, we developed an InstructRestore model that parsed human instructions to achieve region-specific restoration. Our framework allowed users to apply distinct enhancement intensities to different regions and adjust background bokeh effects. By enabling fine-grained control via user instructions, our work advanced research in interactive image restoration and enhancement techniques.

**Limitations.** While InstructRestore offers a baseline for region-customized restoration guided by human instructions, it has several limitations. Currently, it lacks support for instance-level object specification, which requires instance-level masks and captions. Moreover, users are recommended to follow a predefined instruction format, though an off-the-shelf LLM can convert free-form inputs. Furthermore, although our method achieves competitive results, it focuses more on localized customization, while it deserves further exploration of global quality optimization. Reducing the number of inference steps is also worth exploring. Addressing these limitations would boost the applicability and performance of user-instructed image restoration in real world scenarios.

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
