# OpenReview forum: "InstructRestore: Region-Customized Image Restoration with Human Instructions"
_NeurIPS.cc/2025/Conference — NeurIPS 2025 poster_

### Official Review · Reviewer_1V2k · 2025-06-16

**Clarity:** 3
**Significance:** 3
**Originality:** 3
**Rating:** 5
**Confidence:** 4

**Summary:**

This paper investigates a novel problem for real-world general degradation. While existing approaches employ a uniform processing strategy across the entire image, they fail to accommodate users' divergent preferences for different image regions.  The study systematically examines two representative scenarios demanding localized customization, fidelity-perceptual tradeoffs for different semantic regions, and background bokeh control.

To address the aforementioned challenges, this paper proposes a novel interactive restoration framework that accepts natural language instructions indicating the target region and corresponding enhancement scale. It first addresses data scarcity through a generation engine that curates high-quality images and employs Semantic-Sam, Osprey, and Qwen models to produce training triplets (high-resolution images, semantic region masks, and corresponding captions). It then presents an ingenious architecture that achieves region-specific enhancement. While trained using uniform feature fusion strength, during inference, it first parses user-specified regional requirements to generate corresponding masks, then strategically assigns distinct fusion weights to condition features based on masks, achieving spatially variant restoration effects. Experiments verify the system's precise instruction-following capability. Results show it can make targeted adjustments in specified areas while consistently preserving quality in non-target regions.

**Questions:**

In addition to the previous questions, two further clarifications would be helpful:

(1) Are the modulation coefficients bounded (e.g., with predefined min/max limits)? If so, how do these bounds affect the output (e.g., under extreme values)?

(2) The current method focuses on regional control—could the framework inherently support global image-level modulation, or would this require architectural modifications?

**Ethical Concerns:**

["NO or VERY MINOR ethics concerns only"]

**Final Justification:**

Most reviewers think this paper is good and give an accept or weak accept. I also believe this paper is novel, and the instruction-based image restoration is interesting, with great potential value in real-world applications. So I give accept.

**Limitations:**

Yes

**Quality:**

4

**Strengths And Weaknesses:**

**Strengths:**

+ Pioneers the practical yet under-explored task of instruction-guided region-aware restoration, shifting the paradigm from globally uniform processing to spatially adaptive enhancement. This opens new research directions in user-centric image restoration.

+ The approach of decoupling feature fusion strengths in ControlNet during inference to achieve tunable restoration effects is both intellectually interesting and methodologically novel. The experimental results demonstrate impressively effective outcomes.

+ The data engine is technically sound, and the constructed high-quality triplet dataset (530K+ samples) not only supports the current task but also holds significant potential for future low-level vision tasks requiring region-text alignment or extended grounding capabilities.

+ This paper also has a significant practical impact, as it demonstrates compelling use cases that address unmet needs in professional photo editing, i.e. the regional perceptual-fidelity preference and adjustable bokeh preservation. The natural language interface enables intuitive control through verbal instructions, significantly improving workflow efficiency.


**Weakness:**

- This paper demonstrates impressive instruction-guided modulation results, but the underlying mechanism of how the current tuning coefficients achieve such effects could be further analyzed. Notably, in the bokeh effect adjustment, the text prompts fed into the SD backbone differ across instructions. Is this prompt variation inherently linked to the framework's working principle? A deeper discussion would strengthen the theoretical contribution.
- The current framework extracts region captions directly from instructions for the SD backbone input, which prioritizes simplicity. However, prior works like SPIRE show that using more detailed scene descriptions in the backbone's text prompt can enhance global restoration. Would replacing the current region captions with such enriched descriptions further improve performance? A brief ablation study or discussion on this trade-off between simplicity and potential gains would be valuable.

---

> ### Author Rebuttal · Authors · 2025-07-31
>
> We sincerely thank this reviewer for the constructive comments and suggestion. We hope our following point-to-point responses can address this reviewer's concerns.
>
> **[W1]: Clarity of Working Principle.**
>
> We sincerely appreciate this reviewer’s insightful question regarding the interplay between prompt variation and tuning coefficients, which indeed touches the core of our framework’s functionality. As the reviewer astutely observed, the system achieves instruction-guided modulation through a dynamic feature interplay between two key pathways:
>
> *1. SD backbone (text-driven prior).* It receives region-specific captions (e.g., "blurry background") and generates semantically aligned features (e.g., bokeh patterns when instructed).
>
> *2. ControlNet branch (LR-derived reference).* It extracts degradation-aware features from the low-quality input.
>
> The tuning coefficients govern their fusion ratio. Specifically:
>
> *For detail enhancement:* Reducing LR feature weighting allows SD's text-guided features (e.g., "the furry dog") to dominate, synthesizing plausible details.
>
> *For bokeh preservation:* When the SD backbone processes blur-descriptive prompts, lower LR feature contribution enables these blur-oriented features to prevail, while higher LR contribution forces structure recovery.
>
> **[W2]: The Effect of Region Captions**.
>
> We sincerely appreciate this reviewer’s valuable suggestion regarding text prompt enrichment. Following this insightful recommendation, we conducted additional experiments using LLaVA to generate comprehensive scene descriptions for Instruct100Set, replacing our original region-focused captions. As shown in the table, using long captions results in slightly lower reference-based metrics (e.g., PSNR, SSIM) on the full image compared to our standard approach, while the overall performance remains comparable. This is expected, as longer captions with more global descriptions may activate some additional background details, leading to minor changes in the global metrics.
>
> | Method  | Target Area |         |        |       |       | Full Image |         |        |         |        |        |
> |---------|-------------|---------|--------|-------|-------|------------|---------|--------|---------|--------|--------|
> |         | PSNR↑       | SSIM↑   | CLIPIQA↑ | MUSIQ↑ | MANIQA↑ | PSNR↑      | SSIM↑   | LPIPS↓ | CLIPIQA↑ | MUSIQ↑ | MANIQA↑ |
> | Long caption   | 30.52       | 0.8352  | 0.6851 | 67.97 | 0.6129 | 25.52      | 0.6944  | 0.3330 | 0.7276   | 71.96  | 0.6849 |
> | Ours    | 30.55       | 0.8368  | 0.6887 | 68.17 | 0.6137 | 25.65      | 0.6999  | 0.3245 | 0.7278   | 71.95  | 0.6809 |
>
> **[Q1]: The Bound of Modulation Coefficients.**
>
> As analyzed in our responses to [W1], the modulation coefficients govern the feature fusion ratio between the text-guided and LR-derived pathways. The coefficients should maintain a minimum value greater than zero, and our experiments recommended an optimal maximum threshold around 1.5. Exceeding this upper limit tends to introduce artifacts, such as abnormal color saturation effects in the output.
>
> In practice, we do not recommend setting the minimum coefficient below 0.3, as excessively small values cause the output to be dominated by SD backbone features, resulting in images that lack both fine details and structural similarity to the degraded input. It is also worth noting that the visually acceptable range of modulation coefficients can vary across different images: for some images, values up to 1.3 still yield normal results, while for others, artifacts may already appear at this level. Based on our experiments across the entire dataset, 1.5 can be considered a general upper bound for most cases. Therefore, keeping the coefficient within the range from 0.3 to 1.5 ensures a good balance between detail generation and structural fidelity.
>
> **[Q2]: Global Modulation**.
>
> The framework inherently supports global image-level modulation without requiring architectural modifications or retraining. By uniformly applying identical modulation coefficients across both masked and unmasked regions, the system can seamlessly transition from regional to global control.

---

> > ### Comment · Reviewer_1V2k · 2025-08-04
> >
> > The authors have addressed my concerns well. Their framework is well motivated and is a notable strength of the paper. The explanation of the modulation coefficients’ role and the response to the prompt enrichment suggestion has rigorous analysis, with comprehensive experimental results to support their claims.
> >
> > Overall, the instruction-based image restoration is very interesting and has great potential for real-world applications. This work has a solid contribution. Based on the clarity of the responses and reviews from other reviewers, I recommend acceptance of the paper.

---

> > > ### Author Response · Authors · 2025-08-04
> > >
> > > We sincerely thank this reviewer for the recognition on our work and the constructive comments, which are truly encouraging to us! We hope that peers in the community can find our paper helpful, and there will be more works on instruction-based image restoration in the future.
> > >
> > > Authors of paper \#14962

---

### Official Review · Reviewer_DVDK · 2025-06-23

**Clarity:** 3
**Significance:** 3
**Originality:** 3
**Rating:** 4
**Confidence:** 4

**Summary:**

This paper proposes a method to perform region-adjustable image restoration following human instructions. Specifically, it constructs a dataset to support the training and evaluation of this task. Then it develops a ControlNet-like model to enable region-customized image restoration that aligns with user instructions. Extensive experiments are conducted to evaluate the effect of the proposed method.

**Questions:**

1. The authors claim in L44-45 that users generally prioritize perceptual quality for irregular texture regions (e.g., trees), favoring more aggressive detail generation. The statement is not reasonable. For image restoration, the whole image should preserve contetn fidelity and accuracy. However, with generation models, it is harder for irregular texture regions compared to structural elements.
2. Please discuss in detail the effect of the scaling factor \alpha.
3. Please discuss the efficiency of the proposed method and compare with the state-of-the-art methods.
4. It would be better to discuss in detail the effectiveness of the ControlNet features and the restoration mask.

**Ethical Concerns:**

["NO or VERY MINOR ethics concerns only"]

**Final Justification:**

Thanks for the response, which addresses my concerns. I keep my rating and hope the authors could revise the paper according to the rebuttal.

**Limitations:**

yes

**Quality:**

3

**Strengths And Weaknesses:**

Strengths: First, this work advances the investigation of interactive image restoration and enhancement techniques. Second, it constructs a large dataset to support the research of these techniques. Third, it proposes a model to understand user instructions for region-customized restoration.

Weaknesses: The effect of each component in the proposed method is not very clearly demonstrated.

---

> ### Author Rebuttal · Authors · 2025-07-31
>
> We sincerely thank this reviewer for the constructive comments and suggestion. We hope our following point-to-point responses can address this reviewer's concerns.
>
> **[Q1]: The Statement on Irregular Textures.**
>
> Thanks for the comments. We agree that, traditionally, image restoration is defined with a primary focus on content fidelity and accuracy. As mentioned by this reviewer, however, it is indeed challenging to accurately recover pixel-wise details in highly irregular texture regions, including state-of-the-art generative restoration models. This point is well supported by prior literature LDL [1], which indicates that “it is difficult to produce high-fidelity results ... because they have much fine-scale details and suffer from signal aliasing in the degradation process, where most high-frequency components are lost." In these cases, strictly enforcing fidelity often leads to **over-smoothed or visually unconvincing results**, which can negatively impact the user’s perceptual experience.
>
> It is also pointed out in LDL [1] that for such regions: "the pixels are randomly distributed so that the differences between results and ground truth are insensitive to human perception. Therefore, rich details generated can lead to better perceptual quality in these regions."  This confirms that for the Real-ISR task, in certain contexts, perceptual quality is more important to users than strict fidelity. DeSRA [2] also indicates that "fine details in areas with complicated textures are difficult to perceive as artifacts like foliage, hair, and etc, while large pixel-wise differences in areas with smooth or regular textures, such as sea, sky, and buildings, are sensitive to human perception and easy to be seen as artifacts". These findings collectively suggest that the balance between perceptual quality and fidelity should be adaptively considered according to the semantic characteristics of different regions, as user preferences and perceptual sensitivity can vary significantly across image content.
>
> Based on the above observations, our instruction-based region-customized restoration approach is designed to flexibly balance fidelity and perceptual quality: we maintain strict fidelity in structured regions, while allowing more perceptual detail generation in irregular texture areas. We will add these discussions in the revision.
>
> [1] Liang J, Zeng H, Zhang L. Details or artifacts: A locally discriminative learning approach to realistic image super-resolution. CVPR 2022
>
> [2] Xie L, Wang X, Chen X, et al. Desra: detect and delete the artifacts of gan-based real-world super-resolution models. ICML 2023.
>
> **[Q2]: The Effect of Scaling Factor**.
>
> Thanks for the suggestion. The scaling factor plays a central role in our system by controlling the fusion ratio between the SD backbone and the ControlNet branch during inference. By adjusting the scaling factor, the system can flexibly balance between generating new details and preserving fidelity to the degraded input, according to user instructions and regional requirements.
>
> Specifically, SD backbone receives region-specific captions and generates semantically aligned features (text-guided features), while ControlNet branch extracts degradation-aware features from the low-quality input  (LR-derived features).  The scaling factor governs the fusion ratio between these two pathways.
>
> *For detail enhancement*: Reducing the weighting of LR-derived features allows the text-guided features in the SD backbone (e.g., “the furry dog”) to dominate, enabling the synthesis of plausible new details. Please refer to Figure 4 in the main paper for visual effects.
>
> *For bokeh tuning*: When the SD backbone is guided by blur-related prompts, a lower LR feature contribution allows those blur-oriented features to prevail, while a higher LR contribution enforces structure recovery, which may lead to over-restoration of background blur. The visual effects can be seen in Figure 6 of the main paper.
>
> In short, a smaller scaling factor allows the SD backbone to dominate, resulting in richer generated details, while a larger scaling factor makes the output closer to the degraded image, increasing fidelity but reducing generated details.
>
> As for the restoration result, increasing the amount of generated details (i.e., lowering the scaling factor) typically results in lower reference-based metrics (such as PSNR), but higher no-reference perceptual metrics (such as CLIPIQA). This trend can be observed in Table 2 of the main paper: when the fidelity scale (scaling factor) is reduced in the specified region, PSNR and other reference-based metrics decrease, while CLIPIQA and no-reference metrics increase. This demonstrates that the scaling factor provides effective control over the trade-off between fidelity and perceptual quality, allowing users to flexibly adjust restoration outcomes according to their preferences and the semantic context.
>
> **[Q3]: Efficiency**.
>
> Thanks for the suggestion. The table below summarizes the inference steps, inference time (in seconds per image), and parameter count (in billions) for several representative methods. Inference time is measured on 512 × 512 LQ images using a single NVIDIA A100 80G GPU.
>
> |                                 | StableSR | DiffBIR | PASD | OSEDiff | SeeSR | Ours |   |   |   |
> |---------------------------------|----------|---------|------|---------|-------|------|---|---|---|
> | Inference steps                 | 200      | 50      | 20   | 1       | 50    | 20   |   |   |   |
> | Inference time（s）/Image numbers | 10.03    | 2.72    | 2.80 | 0.12    | 4.30  | 4.06 |   |   |   |
> | #Param(B)                       | 1.56     | 1.68    | 2.31 | 1.77    | 2.51  | 1.25 |
>
> Although our method includes a mask decoder, it is relatively lightweight, resulting in a smaller overall parameter count compared to methods like OSEDiff and SeeSR, which require an additional RAM model. However, during inference, our approach generates a mask at each step and performs feature modulation at multiple scales within the SD-UNet. Since our current implementation executes these operations using a for-loop without code optimization, the inference time is a little longer than PASD, which uses the same number of diffusion steps. In our future work, we will (1) optimize our implementation to speed up the inference and (2) explore one-step or few-step inference to further enhance the efficiency of our framework.
>
> **[Q4]: The Effectiveness of ControlNet Features and Restoration Mask**.
>
> We thank the reviewer for prompting a discussion on the effectiveness of the ControlNet features and the restoration mask.
>
> *1. ControlNet features*
>
> As discussed in [Q2], the ControlNet branch extracts degradation-aware features from the low-quality input, providing essential reference information for the restoration process. These features, when fused with the SD backbone via the scaling factor, enable the system to balance between preserving fidelity to the degraded image and generating new details as guided by user instructions.
>
> *2. Restoration mask*
>
> We validate the effectiveness of predicted restoration masks by replacing it with ground-truth (GT) masks during inference and comparing the results on the Instruct100 set under two fidelity settings (0.5 and 1.1).
>
> The table below shows that results with predicted masks are very close to those with GT masks across all metrics in both target and background areas. This demonstrates that the predicted mask is effective to enable accurate region-customized restoration. Furthermore, with either predicted or GT masks, the target region changes with different fidelity scales, while the remaining areas stay stable, confirming precise local control. With a lower fidelity scale (0.5), the difference between predicted and GT masks is slightly larger. This is because, in our multi-step inference framework, a mask is generated at each step, and the predicted masks in the early steps may not accurately cover the target region. As a result, the features of the intended area may still be modulated with a fidelity scale of 1 in the initial steps, leading to higher fidelity metrics compared to using GT masks.
>
> |Fidelity Scale|Target Area| | | | | |Other Area| |
> |-|-|-|-|-|-|-|-|-|
> | |PSNR↑|SSIM↑|LPIPS↓|CLIPIQA↑|MUSIQ↑|MANIQA↑|PSNR↑|SSIM↑|
> |0.5 (predict_mask)|29.71|0.7522|0.1610|0.6801|67.86|0.6108|31.27|0.8949|
> |1.1 (predict_mask)|30.73|0.8649|0.1253|0.6659|66.92|0.5934|31.56|0.9108|
> |0.5 (gt_mask)|29.49|0.7297|0.1663|0.6810|67.76|0.6118|31.32|0.9017|
> |1.1 (gt_mask)|30.74|0.8661|0.1247|0.6663|66.88|0.5930|31.56|0.9107|

---

> > ### Comment · Reviewer_DVDK · 2025-08-05
> >
> > Thanks for the additional clarifications. The rebuttal has addressed my concerns.

---

> > > ### Author Response · Authors · 2025-08-06
> > >
> > > We sincerely appreciate the reviewer's time and insightful feedback! We are pleased and encouraged that our responses satisfactorily addressed your concerns, and we truly appreciate your positive assessment. We hope this paper will serve as a useful resource for peers in the community and help advance research in instruction-based image restoration.
> > >
> > > Authors of Paper #14962

---

> ### Author Response · Authors · 2025-08-04
>
> Dear Reviewer DVDK,
>
> Many thanks for your time in reviewing our paper and your constructive comments. We have submitted the point-to-point responses. We appreciate if you could let us know whether your concerns have been addressed, and we are happy to answer any further questions.
>
> Best regards,
>
> Authors of paper \#14962

---

### Official Review · Reviewer_tpZd · 2025-06-28

**Clarity:** 4
**Significance:** 3
**Originality:** 4
**Rating:** 5
**Confidence:** 5

**Summary:**

The author introduces InstructRestore, a novel framework for region-specific image restoration based on natural language instructions. Besides, the authors construct a large-scale dataset,  containing high-quality images, region masks, and corresponding captions. Compared with existing diffusion-based restoration methods that apply uniform processing across the entire image, InstructRestore enables fine-grained control over different image regions. Experiments show promising results on both general and bokeh-aware restoration tasks.

**Questions:**

- How robust is the model to variations in the instruction format? Some examples or quantitative evaluation would strengthen the claims of instruction-following capability.
- Have the authors tested the model on instructions involving unseen region types or categories not present in the training set?

**Ethical Concerns:**

["NO or VERY MINOR ethics concerns only"]

**Final Justification:**

I appreciate the authors’ thoughtful reply, which has successfully resolved my concerns. After carefully considering the responses and reading the other reviewers’ comments, I have decided to raise my rating from Borderline Accept to Accept, as I believe the paper makes a solid contribution and is suitable for publication.

**Limitations:**

Yes

**Quality:**

3

**Strengths And Weaknesses:**

### Strengths

- The paper addresses a novel and practical problem—region-customized image restoration—through the lens of instruction-following, which is underexplored in existing research.
- The authors contribute a large-scale, high-quality dataset and may benefit related research in instruction-guided vision.
- The proposed framework enables fine-grained, user-controllable restoration, including adjustable fidelity scales for different regions, which is both technically interesting and practically useful.
- The expression is clear.

### Weaknesses

- The instruction format currently relies on structured templates. While effective, it is unclear how the method would respond to natural or slightly varied user expressions.
- The paper lacks ablation studies to isolate the effects of key components such as the mask decoder and the fidelity modulation mechanism.
- The generalization ability of the model to unseen semantic categories or open-domain instructions is not fully explored.

---

> ### Author Rebuttal · Authors · 2025-07-31
>
> We sincerely thank this reviewer for the constructive comments and suggestion. We hope our following point-to-point responses can address this reviewer's concerns.
>
> **[W1] and [Q1]: Variation of Instruction Format**.
>
> We thank the reviewer for raising the concern about the reliance on structured instruction templates. To answer this question, we designed three instruction variants that were not seen during training to evaluate the robustness of our method to more natural or varied user expressions. Specifically, we tested the following variants:
>
> *v1: “make the entire image sharper while make {region expression} clear”;*
>
> *v2: “enhance the clarity of {region expression}”;*
>
> *v3: “I would like {region expression} to be clear”*.
>
> For each variant, we set the foreground fidelity scale to either 0.5 or 1.1, the background to 1, and conducted experiments on the Instruct100 set. The test result is shown in below table.  Notably, v1, which is most similar to the original template, achieves nearly identical results to the original instruction. For v2 and v3, which differ more significantly in phrasing, our method still achieves comparable results, suggesting a certain degree of robustness to instruction variations. All the three instruction variants result in significant changes in the specified region and minimal changes elsewhere, with metrics similar to those under the standard instruction. This suggests that, although the phrasing differs, these instructions can still localize the intended region to some extent. However, we acknowledge that the generated masks are not identical across different instructions, which may cause some result variations.
>
> |Fidelity Scale|Target Area| | | | | |Other Area| |
> |-|-|-|-|-|-|-|-|-|
> | |PSNR↑|SSIM↑|LPIPS↓|CLIPIQA↑|MUSIQ↑|MANIQA↑|PSNR↑|SSIM↑|
> |0.5 (our)|29.71|0.7522|0.1610|0.6801|67.86|0.6108|31.27|0.8949|
> |1.1 (our)|30.73|0.8649|0.1253|0.6659|66.92|0.5934|31.56|0.9108|
> |0.5 (v1)|29.82|0.7636|0.1575|0.6772|67.89|0.6084|31.24|0.8929|
> |1.1 (v1)|30.71|0.8641|0.1239|0.6702|67.19|0.5949|31.55|0.9124|
> |0.5 (v2)|29.92|0.7813|0.1483|0.6580|66.73|0.5914|31.52|0.9050|
> |1.1 (v2)|30.90|0.8814|0.1136|0.6229|64.24|0.5551|31.81|0.9238|
> |0.5 (v3)|29.90|0.7751|0.1506|0.6622|67.09|0.5965|31.40|0.8994|
> |1.1 (v3)|30.84|0.8781|0.1154|0.6361|65.16|0.5654|31.73|0.9210|
>
> **[W2]: Ablation Studies**.
>
> We appreciate the reviewer’s concern about  ablation studies. In our original submission, we did not include experiments that simply remove either the mask decoder or the fidelity modulation mechanism. This is because both modules are essential for achieving region-customized restoration based on user instructions—removing either one would make it impossible to realize the core functionality of localized control.
>
> To address this reviewer’s suggestion and to better analyze the independent contributions of these modules, we have conducted the following additional experiments:
>
> *1. Mask Decoder*
>
> We assessed the mask decoder by replacing its predicted masks with ground truth (GT) masks during inference and comparing restoration results on the Instruct100 set under two fidelity settings (0.5 and 1.1). The table below shows that results with predicted masks are very close to those with GT masks across all metrics in both target and remaining areas. This minimal performance gap demonstrates that our mask decoder generates high-quality masks, enabling accurate region-customized restoration. Furthermore, both predicted and GT masks show that the target region changes much with different fidelity scales, while the remaining areas stay stable, confirming precise local control. Another observation is that at a lower fidelity scale (0.5), where more details are generated in the target region, the difference between predicted and GT masks is slightly larger. This suggests that mask quality is more critical when generating finer details. Nevertheless, the performance gap remains small, further confirming the robustness of our mask decoder.
>
> |Fidelity Scale|Target Area| | | | | |Other Area| |
> |-|-|-|-|-|-|-|-|-|
> | |PSNR↑|SSIM↑|LPIPS↓|CLIPIQA↑|MUSIQ↑|MANIQA↑|PSNR↑|SSIM↑|
> |0.5 (predict_mask)|29.71|0.7522|0.1610|0.6801|67.86|0.6108|31.27|0.8949|
> |1.1 (predict_mask)|30.73|0.8649|0.1253|0.6659|66.92|0.5934|31.56|0.9108|
> |0.5 (gt_mask)|29.49|0.7297|0.1663|0.6810|67.76|0.6118|31.32|0.9017|
> |1.1 (gt_mask)|30.74|0.8661|0.1247|0.6663|66.88|0.5930|31.56|0.9107|
>
> *2. Fidelity Modulation Mechanism*
>
> The fidelity modulation mechanism in our method controls the amount of generated details by adjusting the coefficient for integrating conditional features from ControlNet into the SD backbone. Intuitively, a smaller fidelity coefficient allows the SD backbone to dominate, resulting in richer generated details, while a larger coefficient makes the output closer to the degraded image, increasing fidelity but reducing generated details.
>
> To analyze the effect of the fidelity modulation mechanism, experiments were conducted within our multi-step inference framework (20 steps in total), with the fidelity scale set to 0.6 for the target region and 1 for the remaining area.  We tested the impact of applying the fidelity modulation starting from different inference steps. In our default setting, we apply the fidelity modulation from the very first step (step 0). The results are summarized in the table below. We observe that the later the modulation is applied (i.e., the larger the starting step t), the higher the fidelity metrics and the lower the no-reference quality metrics. This indicates that delaying the application of fidelity modulation will suppress the model’s ability to generate details, and narrow the adjustable range of effects when different fidelity scales are set according to user instructions. Therefore, to ensure sufficient controllability, we apply fidelity modulation from the very beginning of the inference process.
>
> | t steps | PSNR↑  | SSIM↑  | LPIPS↓ | CLIPIQA↑ | MUSIQ↑ | MANIQA↑ |
> |-------|--------|--------|--------|----------|--------|---------|
> | 15   | 30.52  | 0.8454 | 0.1333 | 0.6821   | 67.65  | 0.6055  |
> | 10   | 30.41  | 0.8325 | 0.1371 | 0.6824   | 67.86  | 0.6083  |
> |  5   | 30.28  | 0.8137 | 0.1431 | 0.6866   | 67.95  | 0.6116  |
> |  0 (Ours)| 30.09  | 0.7922 | 0.1511 | 0.6911   | 68.12  | 0.6152  |
>
> **[W3] and [Q2]: Generalization for Unseen Semantic Categories**.
>
> We sincerely appreciate this reviewer's insightful question regarding the model’s generalization to unseen semantic categories and open-domain instructions. Actually, our approach leverages Semantic-SAM to generate open-vocabulary masks without pre-defining a closed-set category list, combined with Osprey for region-specific captioning. This pipeline ensures that the training data encompasses diverse semantic regions beyond conventional category boundaries. For example, as shown in Figure 3 of the Appendix, the specified region is “pagoda”, which is not a predefined category in closed-set semantic segmentation. Nevertheless, our method is still able to localize and adjust this region effectively.
>
> While this design enhances open-domain capability, we fully acknowledge that certain rare or novel categories may still pose challenges due to their limited presence in the training set. For instance, when given uncommon objects such as “astrolabe” or “scaffold” as the specified region, the mask decoder may generate an all-zero mask, making it impossible to localize or adjust the intended area.

---

> ### Comment · Reviewer_tpZd · 2025-08-03
>
> I appreciate the authors’ thoughtful reply, which has successfully resolved my concerns. After carefully considering the responses and reading the other reviewers’ comments, I have decided to raise my rating from Borderline Accept to Accept, as I believe the paper makes a solid contribution and is suitable for publication.

---

> > ### Author Response · Authors · 2025-08-03
> >
> > We sincerely thank this reviewer for the time engaged in reviewing our paper and the valuable comments. Your positive feedback and recognition on our work are very encouraging to us! We hope our paper can be helpful to peers in the community and facilitate the research in instruction-based image restoration.

---

### Official Review · Reviewer_qjkL · 2025-07-02

**Clarity:** 3
**Significance:** 2
**Originality:** 3
**Rating:** 4
**Confidence:** 5

**Summary:**

This paper integrates image super-resolution with the bokeh task, enabling precise restoration of target regions and differentiated background processing based on user instructions. By integrating models like Semantic-SAM, Osprey, and Qwen, it builds an image–mask–text triplet. Using ControlNet, it parses user instructions to generate spatial masks and applies different enhancement scales, achieving fine-grained, end-to-end control over region and intensity.

**Questions:**

1. Different Bokeh scales should be fully matched with the instructions. Apart from Figure 6(a), more experimental results should be provided to demonstrate this point.

2.  What is the purpose of Figure 6(b)? As the parameter S1 decreases (0.6h and 0.4), more details inconsistent with the real images are generated, suggesting that more imagined components are introduced. This seems to reflect an adjustment of the consistency and detail generation capability of the diffusion model-based image super-resolution, rather than enhancing the clarity of the foreground objects.

3. The experimental comparison in Table 4 is unfair. Even though the two parameters of the model have been set, other comparison methods have not been trained on the Bokeh training set, making it an OOD task for them.

**Ethical Concerns:**

["NO or VERY MINOR ethics concerns only"]

**Final Justification:**

This paper is well-written and easy to read. While I raised some questions, such as limited novelty and unfair comparison, during the initial review stage, the authors have effectively addressed most of these concerns in their rebuttal and the subsequent discussion phase. Therefore, I decided to raise my score to a borderline acceptance.

**Limitations:**

yes

**Quality:**

2

**Strengths And Weaknesses:**

Strengths:
1. This work combines image super-resolution with bokeh, showing practical application value in scenarios requiring both clear foregrounds and blurred backgrounds.
2. The constructed triplet dataset is valuable for region-related image editing tasks.
3. The paper is well-written and easy to read.

Weaknesses：
1. The novelty is limited and largely based on combinatory innovation. The integration of natural language instructions with region editing in the image editing domain (e.g., IIR-Net [1]) has already been widely explored. Additionally, there are existing works based on natural language instructions in the image super-resolution field (e.g., SPIRE [2], InstructIR [3], PromptSR [4] ). Furthermore, the use of region masks in this paper has also been studied in the image editing field (e.g., DIFFEDIT [5]). Therefore, the paper does not seem to introduce a new contribution in terms of methodology.

2. The title "Instruction-Customized Image Restoration" is inappropriate. The paper only addresses one task in image restoration (super-resolution) and one task in image editing (Bokeh). Firstly, Bokeh is not an image restoration task but rather belongs to image beautification or editing. Secondly, common image restoration tasks such as dehazing, denoising, low-light enhancement, and deblurring are not explored in this work. Therefore, the task setting in this paper is not appropriate.

3. There is a lack of comparison with recent works, such as image super-resolution methods like InvSR [6], PiSASR [7], and others. Additionally, there is no comparison with the performance of Bokeh methods like drbokeh [8], bokehme [9], etc.

4. In the image super-resolution task, the authors compare their method on the custom-built Instruct100Set rather than widely-used image super-resolution benchmark datasets, which diminishes the persuasiveness of the results. Additionally, the comparison methods lack the latest image super-resolution techniques mentioned earlier. Furthermore, Figures 10 and 11 in the supplementary materials show that the visual quality of the image super-resolution results does not surpass OSEdiff, and the proposed method lacks detail, appearing blurry and smeared.

[1] Zhang, Zhongping, et al. "Text-to-image editing by image information removal." Proceedings of the IEEE/CVF winter conference on applications of computer vision. 2024.

[2] Qi, Chenyang, et al. "Spire: Semantic prompt-driven image restoration." European Conference on Computer Vision. Cham: Springer Nature Switzerland, 2024.

[3] Conde, Marcos V., Gregor Geigle, and Radu Timofte. "Instructir: High-quality image restoration following human instructions." European Conference on Computer Vision. Cham: Springer Nature Switzerland, 2024.

[4] Chen, Zheng, et al. "Image super-resolution with text prompt diffusion." arXiv preprint arXiv:2311.14282 (2023).

[5] Couairon, Guillaume, et al. "Diffedit: Diffusion-based semantic image editing with mask guidance." arXiv preprint arXiv:2210.11427 (2022).

[6] Yue, Zongsheng, Kang Liao, and Chen Change Loy. "Arbitrary-steps image super-resolution via diffusion inversion." Proceedings of the Computer Vision and Pattern Recognition Conference. 2025.

[7] Sun, Lingchen, et al. "Pixel-level and semantic-level adjustable super-resolution: A dual-lora approach." Proceedings of the Computer Vision and Pattern Recognition Conference. 2025.

[8] Sheng, Yichen, et al. "Dr. bokeh: differentiable occlusion-aware bokeh rendering." Proceedings of the IEEE/CVF Conference on Computer Vision and Pattern Recognition. 2024.

[9] Peng, Juewen, et al. "Bokehme: When neural rendering meets classical rendering." Proceedings of the IEEE/CVF conference on computer vision and pattern recognition. 2022.

---

> ### Author Rebuttal · Authors · 2025-07-31
>
> We sincerely  thank the reviewer for the constructive comments and hope our point-to-point responses address the concerns raised.
>
> **[W1]: Methodology Novelty.**
>
> We appreciate this reviewer for the comments and provided literature. However, we believe there are misunderstandings here. Our work is fundamentally different from both image editing and prior instruction-based restoration. Our method tackles the unexplored challenge of region-customized restoration, with unique task formulation and technical implementation.
>
> *1. Difference from Image Editing*
>
> Image editing modifies clean inputs to alter semantics (e.g., scene composition or object attributes), while InstructRestore processes degraded inputs while preserving semantics. The ill-posed nature of restoration implies multiple valid solutions for a degraded input, where users may prefer varying perceptual details across regions. Our goal is to recover content while enabling continuous, intent-driven regional control.
>
> The image editing methods mentioned by this reviewer (e.g., DiffEdit, IIRNet) operate on clean-domain inputs to transform an image by using target text descriptions. Typical approaches first encode the clean image into the noise for diffusion models, then perform denoising using target text prompts. For example, DiffEdit employs DDIM encoding to obtain this noise, then substitutes masked-area noise with text-aligned versions during decoding to alter content. However, these methods cannot be used for restoration. First, the noise-conversion pipeline fails on degraded inputs, hurting output fidelity. Second, unlike editing, our target output shares identical semantic descriptions with the input.
>
> Other editing methods require supervised training with edited image pairs, such as IIRNet mentioned by this reviewer. IIRNet conditions on clean inputs but deliberately corrupts specified regions (via color removal/noise injection) to avoid identity mapping, forcing text-consistent generation only in masked areas. Such editing approaches fundamentally conflict with the restoration task. In contrast to the deliberate corruption strategy, we prioritize both strict semantic consistency and maximal information preservation from degraded inputs.
>
> In summary, image editing fundamentally differs from our task. It requires clean inputs and produces semantically altered outputs by design. Even advanced regional editing techniques are incompatible with restoration objectives. Achieving human-aligned perceptual control for region-specific image restoration is the goal of our work.
>
> *2. Difference from Previous Instruction-Based Restoration*
>
> Existing instruction-based restoration methods (InstructIR, PromptSR, and SPIRE) can only achieve global uniform restoration. In contrast, our work enables region-customized restoration. We need to solve previously unaddressed challenges: (1) precisely localizing user-specified regions in degraded images through instruction parsing, (2) enabling continuous effect modulation within target areas, and (3) guaranteeing artifact-free transitions between modified and unmodified zones. All of these requirements are beyond the scope of existing global instruction-based approaches.
>
> *3. Methodological Innovations*
>
> To address the challenges mentioned above, we first created a large-scale dataset specifically designed for this task, and developed a mask decoder that accurately localizes target regions using both user instructions and the degraded image. We found that a ControlNet-based adaptation enables precise restoration fidelity control by modulating feature integration in Stable Diffusion. Based on this, we selectively adjust feature fusion in target regions, preserving original fidelity elsewhere. Our results show seamless blending without visible edge artifacts..
>
> In summary, our method differs from prior work in three key ways: instruction usage, mask generation, and mask operation. Our novelty lies in the building of the first unified framework for controllable region-level image restoration, which cannot be achieved by existing methods.
>
> **[W2]: Title and Task.**
>
> We thank the reviewer for the feedback regarding the title and task scope of our paper. We respectfully clarify the following points:
>
> *1. On the Use of "Image Restoration" in the Title*
>
> Our work follows the paradigm of recently published papers (e.g., SUPIR[1], SPIRE[2], DreamClear[3], FluxIR[4]) that address real-world image restoration under mixed degradations in the wild. Kindly note that all those works use "image restoration" in their titles, and they report experimental results only on real-world image super-resolution, which is one of the two tasks reported in our paper.
>
> Like these works, our degradation model combines noise, blur, JPEG compression, and downsampling to synthesize real-world distortions. We also evaluate on real-world datasets with complex, entangled and unknown degradations. The title "Region-Customized Image Restoration" aligns with this broader definition of the restoration task adopted by the existing works.
>
> [1] Yu F, et al. Scaling up to excellence: Practicing model scaling for photo-realistic image restoration in the wild. CVPR 2024
>
> [2] Qi C, et al. Spire: Semantic prompt-driven image restoration. ECCV 2024
>
> [3] Ai Y, et al. DreamClear: High-Capacity Real-World Image Restoration with Privacy-Safe Dataset Curation. NeuralPS 2024
>
> [4] Deng J, et al. Acquire and then Adapt: Squeezing out Text-to-Image Model for Image Restoration. CVPR 2025
>
> *2. On the Requirement of Bokeh Preservation in Our Task*
>
> This reviewer may misunderstand our experiments on bokeh-preserved image restoration. Please kindly note that "bokeh preservation" is a requirement in our **region-customized restoration** task. Existing SD-based methods often fail to distinguish bokeh from degraded images, leading to over-restoration of blurred area.  Our approach allows users to specify background blur strength and foreground detail recovery through instructions. It is not an image editing or beautification problem, but a user-instructed image restoration problem.
>
> *3. Concerns on Broader Tasks*
>
> Regarding more restoration tasks such as dehazing, denoising, low-light enhancement, we believe that our data generation pipeline and instruction-based restoration framework can be extended to these tasks, which will have their own unique requirements for localized adjustment. In this work, we focus on the task of real-world image restoration [1-4]. We hope that our work can inspire more researchers to work in this underexplored but valuable direction.
>
> **[W3]: Comparison Baselines.**
>
> *1. Regarding real-world super-resolution methods (InvSR/PiSASR):*
>
> We compare with InvSR/PiSASR in table below. Note that the metrics are computed in the target area. Sorry that we cannot show visual comparisons due to NeurIPS' policy.Actually, like other Real-ISR methods in the main paper, these methods cannot achieve region-level restoration guided by human instructions.The metrics in the table cannot reflect the real performance. We will add the two methods in revision.
>
> | Method | PSNR↑| SSIM↑ | CLIPIQA↑ | MUSIQ↑| MANIQA↑ |
> |-|-|-|-|-|-|
> |PisaSR|30.53|0.8783|0.6648|67.90|0.6028|
> |InvSR|29.89|0.8321|0.6491|67.58|0.6051|
> |Ours|30.55|0.8368|0.6887|68.17|0.6137|
>
> *2. Regarding bokeh rendering methods (drbokeh/bokehme):*
>
> Please kindly note that bokeh rendering methods (e.g., DRBokeh, BokehMe) are architecturally incompatible with our task. They synthetically generate bokeh from clean images with depth maps, while we restore natural bokeh from degraded inputs without depth data. So these approaches could not serve as baselines for our task.
>
> **[W4]: Test-sets and Comparison.**
>
> Kindly note that no existing benchmark suits our novel task. While current SD-based SR methods evaluate on 512×512 crops from RealSR/DRealSR, these lack sufficient semantic diversity for region-specific instructions. Therefore, we created Instruct100Set by strategically selecting semantically rich regions from RealSR, DRealSR and RAIM containing multiple distinct elements, enabling comprehensive evaluation of localized restoration under real-world degradations. Kindly note that Figures 10 and 11 is to show that only our method preserves background blur as intended, unlike others that over-sharpen these regions.
>
> **[Q1]: More Experiments for Bokeh Tuning.**
>
> Following your suggestion, we fixed the foreground fidelity at 0.8 while varying background bokeh fidelity (0.4,0.7,1), where lower values indicate stronger blur. We measured foreground PSNR(fg_psnr), background PSNR(bg_psnr), and background blur strength using the Brenner metric (lower values means stronger blur).
>
> As shown in the below table, there are limited variations in foreground PSNR (0.5 dB) but significant changes in background PSNR (2.52 dB). Moreover, reducing the bokeh fidelity scale lowers Brenner values, confirming increased blur as intended.
>
> |Scale|fg_psnr|bg_psnr|Brenner|
> |-|-|-|-|
> |0.4|29.42|28.80|137.23|
> |0.7|29.87|30.42|150.27|
> |1.0|29.90|31.32|156.44|
>
> **[Q2]: The Purpose of Figure 6(b).**
>
> This figure shows our method's dual capability: controlling background blur restoration while regulating foreground detail generation.
>
> **[Q3]: OOD Training Concern.**
>
> Existing datasets (LSDIR, Unsplash) contain bokeh images, so all methods have seen blurred backgrounds during training. However, severe degradation makes natural bokeh hard to distinguish. Guided by user instructions, our model preserves blur while restoring targets. We also retrained OSEDiff with our bokeh-enriched data. As shown in the table below, it achieves slight background metric improvements over original OSEDiff, but lags behind our method. The background bokeh distortions remain (images omitted due to NeurIPS' policy).
>
> |Method|PSNR↑|SSIM↑|BokehIoU↑|
> |-|-|-|-|
> |OSEDiff (ori data)|29.89|0.8175|0.7990|
> |OSEDiff (bokeh-enriched data)|29.99|0.8208|0.8005|
> |Ours|31.46|0.8462|0.8482|

---

> > ### Comment · Reviewer_qjkL · 2025-08-04
> > **Reply to rebuttal**
> >
> > Thank you to the authors for carefully considering my review comments and providing detailed responses. I believe the authors’ replies have addressed some of my concerns.
> >
> > I hope the authors don’t mind if I raise a few follow-up questions for further clarification.
> >
> > 1. Are the results in Figure 5 and Table 3 fair? If the proposed method uses additional textual prompts, the comparison would not be fair.
> >
> > 2. As I know, the OSEDiff, InvSR, and PiSA-SR are all one-step SR methods, but this method requires tens of inference steps to obtain the result, which is actually not fair, especially considering the non-reference metrics can be largely improved with increased inference steps. Besides, these methods mainly utilize parameter-efficient fine-tuning, such as LoRA, to adapt the diffusion model to SR. However, the proposed method employs a controlnet-like architecture, which may introduce many additional trainable parameters compared to these methods.

---

> > > ### Author Response · Authors · 2025-08-05
> > >
> > > We are happy to know that part of the concerns of this reviewer have been addressed. For the two follow-up questions of this reviewer, we'd like to first clarify that those compared methods in Figure 5 and Table 3 actually cannot solve the problem of region-customized, instruction-based image restoration. We employ them in the experiments just for reference, since there is no other more suitable comparison method available. Please find in the following our more detailed explanation.
> > >
> > >  **[Q1]: Additional textual prompt.**
> > >
> > > Among the compared SD-based methods, except for StableSR and DiffBIR, the remaining methods actually utilize textual prompts as input. Therefore, we understand that the reviewer’s concern about “additional textual prompts” specifically refers to the instruction used in our method.
> > >
> > > Kindly note that our work aims to address the problem of region-customized, instruction-based image restoration. Therefore, human instructions are a MUST for our method. This is fundamentally different from the existing global restoration approaches, including those we compared in Figure 5 and Table 3, which cannot solve this challenging task.
> > >
> > > The results in Table 3 demonstrate that the restoration quality of our method is comparable to existing methods even with a fixed enhancement coefficient across the entire dataset. Figure 5 further illustrates that current restoration methods cannot handle all regions satisfactorily, highlighting the necessity of our newly developed functionality, which allows users to adjust local regions according to their preferences. Our goal is not to achieve state-of-the-art global restoration under a default setting, but is to introduce and solve the task of region-customized restoration. In this sense, it is improper to say that the comparison is “unfair” because the problem settings are inherently different.
> > >
> > > Actually, in terms of network input, our method does not introduce more textual information. Our current instruction format, “make {region caption} clear,” adds only a few extra words beyond the region caption itself. Methods using ControlNet, such as PASD, SeeSR, and SUPIR, also feed a text prompt to the ControlNet branch, which is typically identical to the one input to the SD backbone. SUPIR, in particular, uses even more detailed and longer descriptions. In their actual implementations, the text prompt not only includes the scene description but also appends extra prompt words such as “clean,” “high-resolution,” or “8k.”
> > >
> > >  **[Q2]: Comparison with one-step methods.**
> > >
> > > As discussed in Q1, our work does not aim to achieve state-of-the-art efficiency for global restoration, but rather to introduce and validate a new task that current methods cannot support.
> > >
> > > We mainly compared with multi-step diffusion-based methods like PASD and SUPIR (with the same number of inference steps), as well as SeeSR, StableSR, and DiffBIR (which use even more steps), because multi-step methods have stronger detail generation capability than those one-step methods such as OSEDiff, InvSR, and PiSA-SR. All of these methods, however, may produce regional artifacts that do not align with user preferences.
> > >
> > > Regarding parameter efficiency, our method employs a ControlNet-like architecture, which introduces additional parameters because it generates 2D feature maps for effective mask-based regional control. In contrast, LoRA-based methods learn parameter increments that are applied globally when loaded, making it difficult to achieve localized control. How to design a more elegant regional LoRA mechanism is an interesting direction for future work.
> > >
> > > Overall, our work is the first meaningful exploration of region-customized instruction-based image restoration. We agree that improving the model efficiency and reducing the model parameters are important future directions for this task. Actually, we have tested that reducing the number of diffusion steps from 20 to 10 will achieve very close performance.
> > > | inference steps | PSNR | SSIM | CLIPIQA | MUSIQ | MANIQA |
> > > |------|------|------|------|------|------|
> > > |10|30.63|0.8457|0.7030|68.12|0.6111|
> > > |20|30.55|0.8368|0.6887|68.17| 0.6137|

---

> > > > ### Author Response · Authors · 2025-08-07
> > > >
> > > > Dear Reviewer qjkL,
> > > >
> > > > Thanks for your further questions to our work, and we have posted our explanations. Since the deadline of the reviewer-author discussion period is coming soon, we appreciate if you could let us know whether your further concerns have been addressed.
> > > >
> > > > Best regards,
> > > >
> > > > Authors of paper \#14962

---

> ### Author Response · Authors · 2025-08-04
>
> Dear Reviewer qjkL,
>
> Many thanks for your time in reviewing our paper and your constructive comments. We have submitted the point-to-point responses. We appreciate if you could let us know whether your concerns have been addressed, and we are happy to answer any further questions.
>
> Best regards,
>
> Authors of paper \#14962

---

### Decision · Program_Chairs · 2025-09-17

**Decision:**

Accept (poster)

**Comment:**

The paper presents a practical framework for instruction-guided, region-customized image restoration. Despite initial concerns from one reviewer regarding novelty and baselines, the authors provided thorough and convincing rebuttals, clarifying the distinctions from prior work and addressing evaluation questions. The final decision is to accept the paper, as all reviewers have reached a consensus and recommend acceptance.